# Providing open-label placebos remotely—A randomized controlled trial in allergic rhinitis

**Tobias Kube**[1,2]*, **Verena E. Hofmann**[1], **Julia A. Glombiewski**[1], **Irving Kirsch**[2]

**1** Pain and Psychotherapy Research Lab, University of Koblenz-Landau, Landau, Germany, **2** Program in Placebo Studies, Beth Israel Deaconess Medical Center, Harvard Medical School, Boston, Massachusetts, United States of America

* kube@uni-landau.de

**Data Availability Statement:** All relevant data are within the manuscript and its Supporting information files.

## Abstract

### Background

Placebos can reduce physical symptoms even when provided with full honesty and disclosure. Yet, the precise mechanisms underlying the effects of "open-label placebos" (OLPs) have remained subject of debate. Furthermore, it is unclear whether OLPs are similarly effective when provided remotely, as is sometimes required e.g. in the current COVID-19 pandemic.

### Methods

In a randomized-controlled trial, we examined the effects of OLP plus treatment as usual (TAU) compared to TAU alone on symptom reduction in people with allergic rhinitis ($N = 54$) over the course of two weeks. Due to the COVID-19 pandemic, OLP was provided remotely (i.e. sent via postal service). To investigate the potential influence of the clinical encounter on the effects of OLP, we manipulated the perception of the virtual clinical encounter, both with respect to verbal and nonverbal factors (augmented vs. limited encounter).

### Results

The results of the manipulation check confirmed that the augmented clinical encounter was evaluated more positively than the limited encounter, in terms of perceived warmth of the provider. Participants from all treatment groups showed significant symptom reduction from baseline to two weeks later, but OLP had no incremental effect over TAU. Participants benefitted more from OLP when they did not take any other medication against allergic symptoms than when taking medication on demand. When controlling for baseline symptoms, a significant treatment by encounter interaction was found, pointing to greater symptom improvement in the OLP group when the encounter was augmented, whereas the control group improved more when the encounter was limited.

### Discussion

The study demonstrates that providing OLP and enhancing the encounter remotely is possible, but their effectiveness might be lower in comparison to previous studies relying on

**Funding:** The author(s) received no specific funding for this work.

**Competing interests:** The authors have declared that no competing interests exist.

physical patient-provider interaction. The study raises questions for future research about the potential and challenges of remote placebo studies and virtual clinical encounters. The study has been registered as a clinical trial at ISRCTN (record number: 39018).

## Introduction

Allergic rhinitis, causing symptoms like sneezing, runny nose or itchy eyes, is fairly common in the general population. About 20% of people living in Western countries report symptoms of allergic rhinitis, particularly in the pollen season [1]. Although there are several medications available that help patients deal with their symptoms, research has shown that the placebo response contributes substantially to symptom improvement in allergic rhinitis [2–4]. Until recently, it has been believed that placebo effects in clinical practice require the patients' belief that they are receiving active medication while in fact receiving placebo. However, research has demonstrated that placebos being honestly prescribed to patients (so called "open-label placebos" = OLPs) can lead to symptom reduction in irritable bowel syndrome [5], chronic back pain [6, 7], migraine [8], cancer-related fatigue [9, 10], attention-deficit hyperactivity disorder [11, 12], depression [13, 14], and test anxiety [15]. Recently, it has been shown that OLP also improves symptoms in allergic rhinitis [16, 17].

While there is an increasing body of literature suggesting that OLP can be an effective treatment option, its underlying mechanisms of action have remained insufficiently explained [18–20]. In addition to learning mechanisms and expectancies, the role of the provision of a convincing rationale has received particular attention in previous research. One study in experimentally induced pain found that OLP with a rationale was more effective than OLP without a rationale, while OLP with a rationale was not different from deceptive placebo [21], pointing to the importance of a plausible rationale. In allergic rhinitis, however, Schäfer et al. [17] failed to replicate this effect. The authors examined 46 people with allergic rhinitis to investigate the effects of treatment as usual (TAU) vs. TAU + OLP and rationale vs. no rationale. They found that participants receiving OLP reported greater symptom improvement than the control group, but this effect was not dependent on the provision of a convincing rationale [17]. The present study built on that prior work and aimed to examine the clinical encounter as an additional factor that might contribute to the effects of OLP in allergic rhinitis.

Previous research on the role of the clinical encounter in placebo studies has shown that the placebo response in antidepressant clinical trials increases with the number of clinic visits [22]. Similarly, in a study on sham acupuncture in patients with irritable bowel syndrome, Kaptchuk et al. [23] demonstrated that an augmented clinical encounter (in terms of expressing interest in the patient's symptom experience and life situation, active listening, and showing empathy) elicited a larger placebo response than a limited clinical encounter, while the latter was still superior to a waitlist control group. In an experimental study focusing on allergic reactions, Howe et al. [24] varied the provider's social behavior in terms of high vs. low warmth and high vs. low competence to examine the influence of the clinical encounter on the placebo effect. The authors found that the combination of high warmth and high competence enhanced the placebo response by fostering more positive expectancies in participants. Yet, Howe et al. used deceptive placebos and to our knowledge, no study has investigated whether these findings underscoring the importance of the clinical encounter also apply to the administration of OLP. Therefore, the goal of the present study was to vary both the treatment provided (TAU vs. TAU + OLP) and the clinical encounter (augmented vs. limited) in allergic

rhinitis. This 2 by 2 design would allow us to examine not only the main effects of OLP and the clinical encounter, but also their interaction. Tying in with previous work, we hypothesized that OLP + TAU would lead to greater symptom improvement than TAU alone, and that this effect would be particularly pronounced in the case of an augmented clinical encounter.

Originally, we had planned to test these hypotheses in a study with physical contact between the provider and the patient. Due to the COVID-19 pandemic, however, this original plan was thwarted. As the recruitment of our study relied on the pollen season (usually lasting from ~March to August), we had to choose between either putting the study on hold for at least a year or quickly come up with a concept to conduct the study remotely. We opted for the latter (as described in detail in the methods section). To our knowledge, our study is thus the first to examine the effects of OLP remotely; therefore, in addition to testing the aforementioned hypotheses, a further goal of the present study was to examine whether providing OLPs remotely is feasible and similarly effective as compared to previous studies with physical contact between patients and the provider. Relatedly, the present study also aimed at investigating whether the variation of a clinical encounter (augmented vs. limited) is possible when the encounter takes place virtually.

## Materials and methods

The study design was a randomized-controlled trial, with two factors being examined: treatment (OLP+TAU vs. TAU) and clinical encounter (augmented vs. limited). The study was approved by the Institutional Review Board of the University of Koblenz-Landau (reference number 2020_236) and was conducted in accordance with ethical standards as laid down in the 1964 Declaration of Helsinki and its later amendments. All participants gave informed consent. The study protocol was pre-registered on AsPredicted.org: https://aspredicted.org/t34su.pdf. Furthermore, the study has been registered as a clinical trial at ISRCTN (record number: 39018); yet, this registration has been done after the completion of the trial, since it had been pre-registered on AsPredicted already, where we found the format to pre-specify the hypotheses and planned analyses to be more intuitive. The authors confirm that there are no further trials relating to this drug/intervention ongoing.

### Participants

The sample size was determined via an a-priori power analysis. We estimated the expected effect size based on the results provided by Schäfer et al. [16, 17] who found large effects of OLP on symptom improvement in allergic rhinitis. Accordingly, we expected a large effect ($f = 0.4$) of OLP (vs. TAU) on symptom improvement, and the power analysis using G*Power for an analysis of variance with fixed effects, special, main effects and interactions indicated a required sample size of at least 52 people (alpha:.05; power:.80). Participants were recruited via email lists, posters in public spaces, newspaper announcements, and social media. The inclusion criteria were: diagnosed allergic rhinitis; at least 18 years old; and sufficient German language skills. Similar to previous studies [16, 17], the exclusion criteria were: pregnancy; diabetes; any mental or neurological illness; and lactose intolerance (because the placebo tablets contained lactose). Importantly, no restrictions regarding the participants' intake of their normal medication were made, but participants were asked not to change their medication (or dosages) during the study period. Fig 1 shows the participants' flow in a CONSORT diagram.

### Procedure

To make sure that our results are well comparable to the previous studies by Schäfer et al. [16, 17], we aimed to keep the procedure as similar as possible to their protocols. Participants who

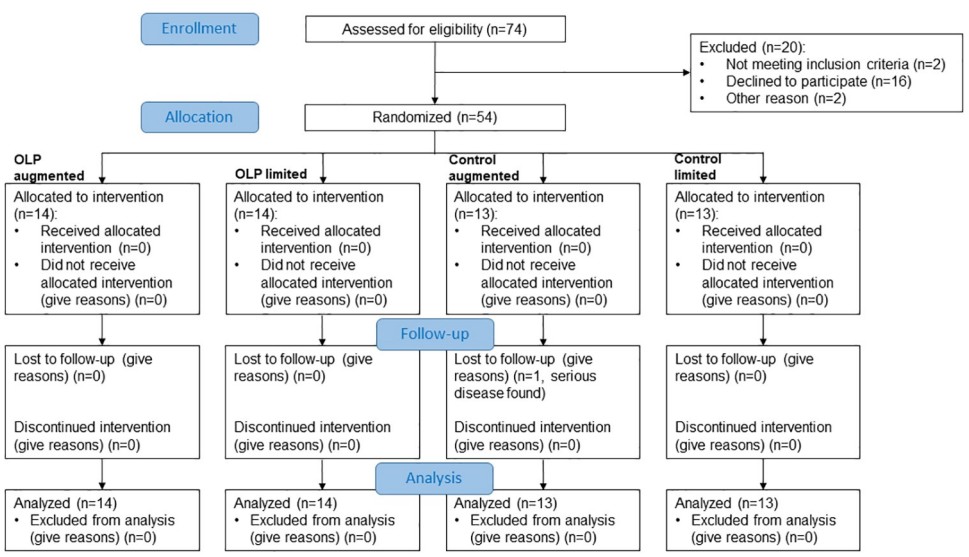

**Fig 1. CONSORT diagram of the participants' flow.**

were interested in the study contacted the study team that checked whether they were eligible for the study. If participants were included in the study, an appointment for the first study visit was made (T1). Due to the COVID-19 pandemic, the appointment took place online. To this end, we used the video platform www.arztkonsultation.de, which is widely recommended and used in Germany, as it meets strict criteria of data safety. This was important for our study since we aimed to ask participants for several pieces of medical and personal information. Prior to the first virtual encounter, it was determined by randomization whether participants would receive the augmented or the limited clinical encounter. As part of the first virtual study visit, the provider—a female psychology master student—explained the administration of OLP to the participants. (Of note, since previous research found that nocebo effects were greater when subjects were in the presence of a person of the same sex than when in the presence of a person of the opposite sex in some conditions [25], we checked whether participants' gender influences the results. It did not, as our analyses indicate, which is why we do not refer to this in further detail in the results section below). In doing so, we closely adhered to previous studies providing OLP [5, 6, 9, 13, 15–17, 21, 26], while stressing in particular that OLPs have been shown to be effective in reducing symptoms of allergic rhinitis [16, 17]. After that explanation, all participants completed questionnaires to rate their expectancies for placebo treatment and the degree to which they felt informed about placebos. Additionally, participants were provided with some open-ended questions asking for their knowledge about placebos, in order to make sure that they understood the explanations about placebos provided beforehand. Furthermore, participants completed the questionnaires assessing their current allergic symptoms. All these measures were completed online using they survey platform www.soscisurvey.de.

After completing the above-mentioned assessments, the provider randomized participants to either the TAU or the TAU+OLP group. If participants were randomized to TAU+OLP (subsequently referred to as "OLP" if not otherwise mentioned), participants were sent the placebo pills via postal service. Due to the COVID-19 pandemic, providing the placebos in a physical encounter was not possible, which is why we developed the idea of sending the placebos via postal service. The placebo tablets were produced by a local pharmacy, according to the ingredients mentioned by Schäfer et al. [16, 17]. That is, placebo tablets were white, round, about 4 mm and contained sugar, lactose, wheat- and cornstarch, lactose monohydrate,

cellulose-powder, magnesium stearate, microcrystalline cellulose, and glucose syrup. The placebos were in a small glass container, which was sent to the participants in a padded envelope. The container had a label on which the logo of the University of Koblenz-Landau was displayed, supplemented by the headline "Placebo Tablets". Participants were asked to swallow the placebos, not to chew or suck them, twice a day (one tablet in the morning and another one in the evening). As with Schäfer et al., participants took the placebos for 14 days. In addition to OLP, participants were allowed to continue to take their regular medication (if there was any), but were asked not to change their medication until the second study visit. Participants from the TAU group (subsequently referred to as "control group") did not receive placebos after the first virtual study visit. With respect to their regular treatment, they received the same information as participants from the OLP group.

To make sure that all participants could take the placebos for 14 days, we scheduled the second virtual appointment about 17 days after the first appointment, taking into account that the postal service would take 2–3 days to deliver the placebos. The second virtual appointment (T2) took place again via www.arztkonsultation.de. At this appointment, the provider's behavior was not manipulated, i.e., it was the same for all groups. At the beginning of the second appointment, the provider asked participants to complete the follow-up questionnaire for their allergic symptoms. Subsequently, using semi-structured interview questions, the provider asked participants how they experienced taking the placebo and whether they noticed any beneficial or adverse effects.

If participants were randomized to the control group at the first appointment, they were offered the possibility of receiving the placebos after the second appointment ("switch-over"). Of 26 participants, 16 persons expressed the wish to receive the placebos, accordingly. Regarding the intake of the placebos, participants received the same information as participants from the OLP group at the first appointment. Participants from the control group who wished to take placebos, received an additional third virtual appointment, again ~17 days later (T3). At this appointment, they completed the symptom questionnaires and the provider asked for beneficial and adverse effects of the placebos, as described above. For all other participants, the study was completed at the T2 appointment. Data were collected between April 20th 2020 and August 13th 2020. Fig 2 illustrates the procedure of the study.

## Variation of the clinical encounter

The two variations of the clinical encounter followed a written protocol and comprised several verbal and non-verbal aspects as well as contextual factors, as presented in Table 1. To design the two encounter styles, we carefully considered previous work by Kaptchuk et al. [23] and Howe et al. [24]. Our overall goal was that the provider in the limited encounter behaves in a relatively neutral and slightly distant way, with a particular focus on the delivery of a standardized procedure. The provider did not intentionally behave in an unfriendly manner, though; rather, the limited encounter was designed to resemble a relatively short, un-personalized encounter as typical in some medical settings. In the augmented condition, on the other hand, the provider was instructed to behave in a very warm, understanding, and empathic manner. Specifically, with reference to the work by Howe et al. [24], we particularly focused on the manipulation of the provider's warmth, while not varying the provider's competence. For the augmented condition, we also considered the study by van Osch et al. [27] who found that positive affect-oriented communication (expressing warmth and empathy) reduced anxiety, negative mood, and increased satisfaction in people with menstrual pain. As described below, we examined in a pre-test whether the two encounter styles were perceived differently.

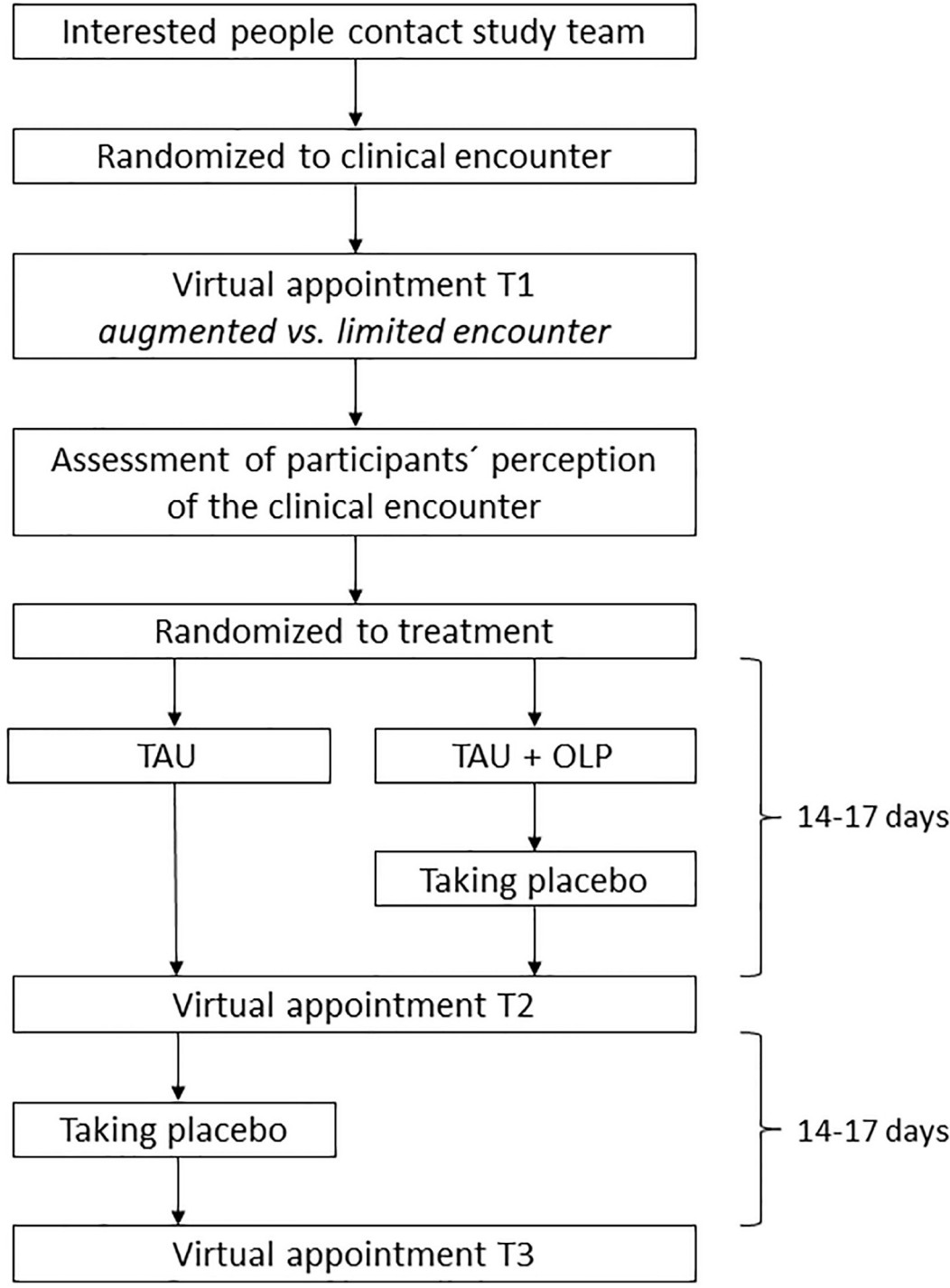

**Fig 2. Illustration of the procedure of the present study.**

### Randomization and blinding

Prior to the first virtual appointment, participants were randomly allocated to one of the two clinical encounter styles (augmented vs. limited) using a computer-generated randomization sequence. At the end of the first virtual encounter, the provider randomized participants to

Table 1. Variation of the clinical encounter.

| Factor | Augmented encounter | Limited encounter |
| --- | --- | --- |
| **Verbal communication** | • Personal introduction; ask for name and introduce self<br>• Room for questions and feedback<br>• Patient-centered, friendly, and warm language; avoid technical jargon or explain it<br>• Ask questions about both symptoms and the psychosocial living situation<br>• Appreciate previous attempts to reduce symptoms<br>• Express gratefulness for participation | • Short introduction; do not ask for name or introduce self<br>• No room for questions and feedback<br>• Encounter like a standardized interview, with the focus on the procedure (not the person)<br>• Technical, matter-of-fact language |
| **Non-verbal communication** | • Much eye contact<br>• Open, facing body posture<br>• Confirming and validating gestures and facial expressions; friendly, smiling facial expression<br>• Patient-centered encounter; warm and caring atmosphere | • Little eye contact, complete questionnaires while talking to the patient<br>• Little gestures and facial expressions (neutral facial expression)<br>• Procedure-centered atmosphere; little warmth and caring<br>• Provider occasionally types something on keyboard |
| **Environmental/context factors** | • In the background: bookshelf, personal items (private photographs) | • No personal items in the background; background shows only a clean white wall |

either OLP or the control group by opening a concealed container (visible to the participants). Thus, neither the provider nor the participant was blinded with respect to the participants' treatment allocation. Participants were not aware, however, of their allocation in terms of the clinical encounter style, since the variation of this factor was disclosed only at the end of the study.

## Measures

**Allergic symptoms.** The primary outcome of the present study was self-reported allergic symptoms. Allergic symptoms were assessed with the Combined Symptom Medication Score (CSMS). This 6-item scale assesses allergy symptoms relating to the nose (four items, e.g., itchy nose) and the eyes (two items, e.g., teary eyes). Each item (reflecting a particular symptom) is rated on a 4-point Likert scale, indicating the severity of symptoms (from 0 = "no symptoms" to 3 = "severe symptoms"). The CSMS distinguishes between currently experienced symptoms, symptoms experienced in the last 12 hours, and symptoms in the last two weeks. In the current study, we focused on the latter as we expected an effect of the placebo treatment on symptoms over the course of two weeks. In addition to the assessment of symptoms, the CSMS can also be used to compute a medication score; however, as participants in the current study were required not to change their medication during the study period (resulting in a constant that would be added to the symptom score), the medication score was not used here and the analyses for symptom improvement in the present study are based on the symptom score only. The CSMS has been recommended by expert consensus as the primary endpoint measure for studies on allergic rhinitis by regulatory authorities in the USA and Europe [28], which is why we used this measure instead of the self-developed and un-validated measure used by Schäfer et al. [16, 17]. Importantly, according to expert consensus [28], the CSMS assesses only eye-related and nose-related symptoms in seasonal allergic conditions, whereas Schäfer et al. [16, 17] also focused on additional symptoms that have been recommended not to be assessed any longer (e.g., breathing, mouth, skin). At baseline, Cronbach's alpha of the CSMS was $\alpha = .79$; at the second appointment two weeks later, Cronbach's alpha was $\alpha = .78$.

**Impairment.** The secondary outcome of the present study was an adapted version of the Pain Disability Index (PDI), which is widely used as a measure of impairment caused by

physical symptoms in behavioral medicine [29]. The PDI comprises seven items to assess the extent to which people feel impaired by their symptoms (such as allergic symptoms) in various areas of life (e.g., household, work, social relationships). Each item is rated on an 11-point Likert scale ranging from 0 ("not at all impaired") to 10 ("completely impaired"). In the present study, Cronbach's alpha of the PDI was $\alpha$ = .86 at baseline and $\alpha$ = .85 at the second appointment two weeks later.

**Perception of the clinical encounter.** To assess participants' perception of the clinical encounter in terms of warmth and competence of the provider, we used an extended version of the Health Screening Experience Questionnaire developed by Howe et al. [24]. The original questionnaire by Howe et al. comprised ten items, and we added another eleven items following the same construction principles. Thus, the questionnaire used in the present study comprised 21 items, fifteen of which assess the warmth of the provider in the encounter and another six items refer to the provider's competence (as presented in the S1 Table). Of note, warmth was assessed with more items, as the main focus of our variation of the clinical encounter was on that aspect, while we did not manipulate the perceived competence of the provider; accordingly, we aimed to assess perceived warmth as precisely as possible, resulting in more items for that subscale. Each item was rated on a 5-point Likert scale ranging from 1 ("I totally disagree") to 5 ("I totally agree"). The extended clinical encounter scale was validated in a pilot study as described below. In the pilot study, the extended scale showed very good psychometric properties: Cronbach's alpha of the warmth subscale was $\alpha$ = .88, and for the competence subscale, $\alpha$ was.90. In the main study reported here, Cronbach's alpha of the warmth subscale was $\alpha$ = .90 and $\alpha$ = .80 for the competence subscale.

**Treatment expectations.** To assess participants' expectations regarding the placebo treatment, we used the treatment expectancy scale [26] as presented in the supplement. This 5-item scale was developed to assess the degree to which participants expect to benefit from a placebo treatment (e.g., "I am confident that the placebo pills will reduce my symptoms"). Each item is rated on a 5-point Likert scale ranging from 1 ("I totally disagree") to 5 ("I totally agree"). Cronbach's alpha of the treatment expectancy scale in the present study was $\alpha$ = .87. The expectancy scale was administered after the rationale for OLP was explained, but before randomization to OLP or TAU.

**Feeling informed about placebos and knowledge about placebos.** Similar to previous studies [21, 26], we asked participants to what extent they felt informed about placebos, using a brief three-item scale ("I feel well informed about placebos and placebo effects"; "I can explain in my own words what placebos are"; "I know how I am supposed to take the placebos"), rated on a 5-point Likert scale ranging from (1) "I totally disagree" to (5) "I totally agree". In addition, we assessed participants' knowledge about placebos and placebo effects to make sure that the information provided by the provider was understood correctly. For this purpose, we asked participants six questions about placebos (e.g., "For what conditions have placebos been shown to be effective?"), as listed in the supplement. Participants were asked to enter their answers in open text fields.

**Medication use.** We assessed whether participants took any medication against their allergic symptoms. Participants were asked to choose one of three options: 1) "I regularly take medication against my allergic symptoms"; 2) "I take medication against my allergic symptoms on demand"; 3) "I don't take any medication against my allergic symptoms".

**Other measures.** Sociodemographic basic variables, including age, sex, and education level were assessed using a brief self-report questionnaire. In addition, as the conduction of the study coincided with the COVID-19 pandemic, we also asked participants whether they were concerned about the possibility that their symptoms could be related to COVID-19, or whether they believed their symptoms to be unrelated to COVID-19.

## Pilot study

Prior to the beginning of the main study, we conducted a pilot study to investigate whether the two variations of the clinical encounter as described above are perceived differently. Specifically, we tested the hypothesis that the augmented clinical encounter is perceived more positively than the limited clinical encounter, particularly in terms of perceived warmth of the provider. To this end, we recorded film clips showing the provider (the same as in the main clinical trial reported below) in a clinical encounter with a (female or male, according to randomization) patient with allergic rhinitis. The provider acted according to a script developed beforehand, covering the features of the two versions of the clinical encounter as described above. In terms of contents of the conversation, we aimed at keeping the encounter in the pilot study as similar as possible to the later encounter in the main study. The video clips were presented in a brief online survey, and participants were asked to rate the provider's behavior in terms of perceived warmth and competence. In addition to the variation of the clinical encounter in terms of warmth and competence, we also manipulated whether the provider informed the patient about placebos within the clinical encounter vs. participants received an information text on placebos (covering the same content of information) after the encounter. We did so because we wanted to rule out the possibility that the augmented clinical encounter is perceived more positively only because it comprised more/different information about placebos. Thus, we examined the following conditions in the pilot study: augmented with information on placebo as part of the encounter vs. augmented with information on placebo separate from the encounter vs. limited with information on placebo as part of the encounter vs. limited with information on placebo separate from the encounter.

In the pilot study, we examined 63 individuals with allergic rhinitis (age 18 to 67; $M = 26.63$ years; 49.2% female), who reported to have allergic symptoms for an average of 12.2 years. The results indicated that the augmented clinical encounter was perceived more positively than the limited clinical encounter in terms of perceived warmth of the provider, $F(1, 59) = 130.730$; $p < .001$; $d = 2.699$, reflecting a very large effect. Also, the provider was perceived as more competent in the augmented condition than in the limited condition, $F(1, 59) = 7.188$; $p = .010$; $d = 0.780$, reflecting a large effect. The correlation between perceived warmth and perceived competence was $r = .600$ ($p < .001$). Furthermore, the results of the pilot study indicated that the clinical encounter was perceived more positively when information about placebos was provided as part of the discussion (as opposed to in written form after the encounter), $F(1, 59) = 7.943$; $p = .007$; $d = 0.455$, but the way of presenting information on placebo did not interact with the augmented vs. limited clinical encounter, $F(1, 59) = 3.645$; $p = .061$; $\eta^2_p = .058$. In sum, the results of this pilot study confirmed that our variation of the clinical encounter indeed led to different perceptions of it, which enabled us to investigate in the main clinical trial whether it would also differentially affect the response to open-label placebo.

## Statistical analyses

First, we conducted data screening according to the suggestions made by Tabachnick and Fidell [30], and tested the assumptions of analyses of variance (ANOVAs). In terms of intention-to-treat analyses, we estimated the missing values of the person who dropped out using the expectation maximization procedure according to methodological recommendations [30, 31]. Performing ANOVA and $\chi^2$-tests, we examined whether the groups differed at baseline in any clinical or sociodemographic variables. Also, we examined in two separate 2 (Treatment: OLP vs. control group) by 2 (Encounter: augmented vs. limited) ANOVAs whether the groups differed in their treatment expectancies and the extent to which they felt informed about placebos. In terms of a manipulation check, we subsequently conducted a t-test to examine whether

the augmented and the limited clinical encounter differed in perceived warmth and competence. In the main analyses, we considered change in allergic symptoms as the primary outcome. Herein, we first performed a 2 (Time: baseline vs. 2-week follow-up) by 2 (Treatment: OLP vs. control group) by 2 (Encounter: augmented vs. limited) mixed ANOVA with allergic symptoms as the dependent variable. When testing main and interaction effects, the following procedure was followed: If the interaction effect was significant, we provided an interpretation of the results, but did not test main effects because the tests for main effects are uninteresting in light of significant interactions. If interaction effects were non-significant, we dropped the interaction effects from the model and tested the main effects. Subsequently, we performed an analysis of covariance (ANCOVA) with the change score in symptom improvement (T1-T2) to take baseline symptoms into account. Similarly, we controlled our main analyses for medication use. Afterwards, we repeated the main analyses for the secondary endpoint, that is, changes in subjective impairment by allergic symptoms. Finally, using qualitative analyses, we examined the occurrence of adverse events and participants' overall feedback on the study. Type-1 error levels were set at 5%. All analyses were conducted using IBM SPSS Statistics Version 25.

## Results

### Sample characteristics

From 74 people screened for eligibility, 54 participants agreed to participate in the study and were randomized to one of four groups (OLP augmented vs. OLP limited vs. control group augmented vs. control group limited). In the entire sample, the mean age was 31.3 years and 68.5% of the participants were female. Regarding current medication against allergic symptoms, 33.3% reported to take medication regularly, while 44.4% indicated that they took medication on demand and another 22.2% did not take any medication against allergic symptoms. The sample characteristics for the four groups separately are detailed in Table 2. Of note, the OLP group was significantly younger than the control group, $F(1, 50) = 6.781$; $p = .012$; $\eta^2_p = .119$. To control for these age differences, we added age as a covariate in all analyses reported below to see whether it affects the results. Doing so revealed that age did not influence the pattern of the results, which is why we do not explicitly refer to this again in the main analyses presented below. Regarding all other variables, there were no significant baseline differences between the groups.

### Expectancies and feeling informed about placebos

A 2x2 (Treatment by Encounter) ANOVA indicated that the two treatment groups did not differ in their treatment expectancies, $F(1, 50) = 0.225$; $p = .637$; $\eta^2_p = .004$, and the extent to which they felt informed about placebos, $F(1, 50) < 0.001$; $p = .991$; $\eta^2_p < .001$. Similarly, people from the augmented vs. limited encounter did not differ in their expectancies $F(1, 50) = 0.035$; $p = .853$; $\eta^2_p = .001$, and the degree to which they felt informed about placebos, $F(1, 50) = 0.965$; $p = .331$; $\eta^2_p = .019$. Also, there was no Treatment by Encounter interaction for expectancies ($F(1, 50) = 1.338$; $p = .253$; $\eta^2_p = .026$) and feeling informed about placebos ($F(1, 50) = 0.965$; $p = .331$; $\eta^2_p = .019$). As these measures were rated prior to randomization, no group differences were to be expected here. Moreover, expectancies ($F(1, 50) = 2.700$; $p = .107$) and feeling informed about placebos ($F(1, 50) = 0.036$; $p = .851$) were not found to be significant moderator variables in the main analyses on symptom change reported below.

**Table 2. Sample characteristics.**

| Variable | OLP-augmented (n = 14) | OLP-limited (n = 14) | TAU-augmented (n = 13) | TAU-limited (n = 13) | Group differences |
|---|---|---|---|---|---|
| Mean age in years (SD) | 26.8 (10.1) | 27.1 (11.0) | 32.2 (11.4) | 39.8 (17.5) | $F(3, 50) = 3.014$; $p = .039$; $\eta^2 = .153$ |
| % female | 64.3 | 64.3 | 69.2 | 76.9 | $\chi^2 = 0.661$; $p = .882$ |
| Educational degree, % | | | | | |
| Primary education | 0 | 0 | 7.7 | 15.4 | $\chi^2 = 9.595$; |
| High school | 57.1 | 71.4 | 30.8 | 61.5 | $p = .384$ |
| University degree | 42.9 | 28.6 | 61.5 | 33.1 | |
| Employment status, % | | | | | |
| University student | 78.6 | 64.3 | 53.8 | 46.2 | $\chi^2 = 16.379$; |
| Employed | 14.3 | 28.6 | 46.2 | 15.4 | $p = .174$ |
| Self-employed | 7.1 | 0 | 0 | 23.1 | |
| Other | 0 | 7.1 | 0 | 15.4 | |
| Current allergic medication, % | | | | | |
| Regular intake | 42.9 | 28.6 | 30.8 | 30.8 | $\chi^2 = 2.258$; $p = .894$ |
| On demand | 42.9 | 42.9 | 38.5 | 53.8 | |
| No medication | 14.2 | 28.6 | 30.8 | 15.4 | |
| Allergic symptoms at baseline, M (SD)[1] | 15.1 (4.8) | 16.2 (2.9) | 15.6 (4.7) | 15.9 (3.4) | $F(3, 50) = 0.181$; $p = .909$; $\eta^2 = .011$ |
| Impairment at baseline, M (SD)[2] | 25.5 (14.0) | 25.5 (11.7) | 25.0 (12.4) | 27.6 (12.2) | $F(3, 50) = 0.112$; $p = .953$; $\eta^2 = .007$ |

*Note*: OLP = open-label placebo; TAU = treatment as usual

[1] Total score ranges from 0 to 18

[2] Total score ranges from 0 to 70

## Manipulation check

The descriptive values of the manipulation check are presented in Table 3. As hypothesized, participants from the augmented clinical encounter condition rated the provider's warmth as significantly higher than participants from the limited encounter group, $t(52) = -2.276$; $p = .027$; $d = .619$, reflecting a medium effect. The two encounter styles did not differ, however, in perceived competence of the provider, $t(52) = 0.329$; $p = .743$; $d = .089$. Thus, the manipulation was effective in manipulating the perceived warmth of the provider, although the effect size was considerably lower than in the pilot study. The correlation between perceived warmth and perceived competence was $r = .689$ ($p < .001$).

## Primary endpoint: Change in allergic symptoms

The Time by Treatment by Encounter ANOVA with allergic symptoms as the dependent variable indicated no significant interactions. The main effect of Time was significant,

**Table 3. Descriptive values of the manipulation check.**

| | Augmented encounter (n = 27) | Limited encounter (n = 27) | Group differences |
|---|---|---|---|
| Perceived warmth, M (SD) | 4.79 (0.39) | 4.47 (0.59) | $t(52) = 2.276$; p = .027; d = .619 |
| Perceived competence, M (SD) | 4.77 (0.38) | 4.80 (0.30) | $t(52) = 0.329$; p = .743; d = .089 |

*Note*: Both perceived warmth and competence were assessed with a 7-point Likert scale ranging from 1 to 7, with higher values a more positive perception of the encounter.

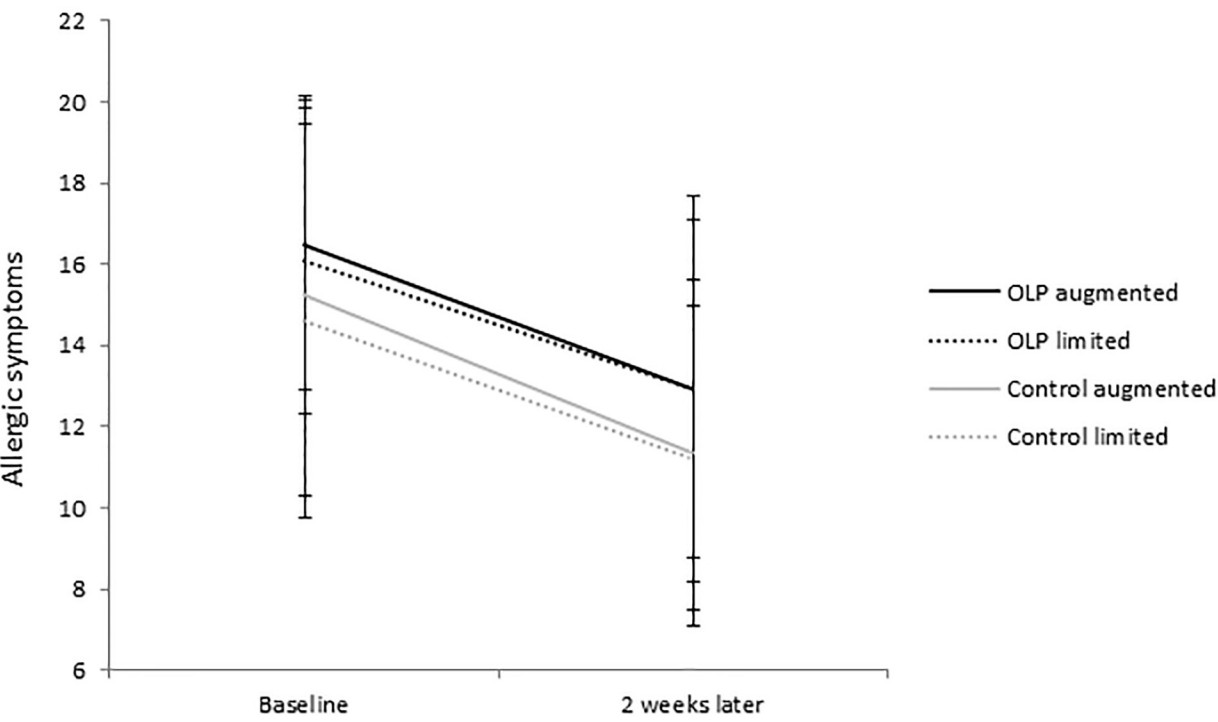

**Fig 3. Change in allergic symptoms in the open-label placebo and the control group.**

$F(1, 50) = 17.691$; $p < .001$; $\eta^2_p = .261$, with overall higher symptom severity at baseline than at follow-up. The main effects of Treatment ($F(1, 50) = 0.089$; $p = .767$; $\eta^2_p = .002$) and Encounter ($F(1, 50) = 0.685$; $p = .412$; $\eta^2_p = .014$) were not significant. The results of this analysis are depicted in Fig 3 and the corresponding descriptive values are presented in Table 4.

When examining changes in nose-related allergic symptoms and eye-related symptoms separately, the pattern of results (that is, the significance of the aforementioned effects and their effect sizes) did not change significantly.

**Controlling for baseline symptoms.** To account for regression artifacts in symptom improvement, we performed an ANCOVA with the change score of allergic symptoms (T1-T2) as the dependent variable and baseline symptoms as the covariate, in line with methodological considerations [32, 33], while again examining the (main and interaction) effects of the factors Treatment and Encounter (note that this ANCOVA yields the same results, in terms of inferential statistics, as an ANCOVA using the post-treatment values as the dependent variable instead of the change score). The ANCOVA indicated a significant Treatment by Encounter interaction, $F(1, 49) = 5.454$; $p = .024$; $\eta_p^2 = .100$, pointing to greater symptom improvement for OLP when the encounter was augmented (adj. $M = 2.61$; $SE = .89$), whereas participants from the control group benefitted more from the limited encounter (adj. $M = -1.63$; $SE = .92$), reflecting a medium effect, $d = .667$. The results of this ANCOVA are illustrated in Fig 4.

**Table 4. Descriptive values of the main analyses regarding change in symptoms and impairment.**

| Variable | OLP-augmented (n = 14) | OLP-limited (n = 14) | TAU-augmented (n = 13) | TAU-limited (n = 13) |
|---|---|---|---|---|
| Allergic symptoms at T1, M (SD)[1] | 15.1 (4.8) | 16.2 (2.9) | 15.6 (4.7) | 15.9 (3.4) |
| Allergic symptoms at T2, M (SD) | 12.0 (3.0) | 14.9 (3.8) | 13.6 (3.5) | 12.1 (3.3) |
| Allergic symptoms at T3, M (SD) | - | - | 12.2 (4.3) | 11.6 (4.5) |
| Impairment at T1, M (SD)[2] | 25.5 (14.0) | 25.5 (11.7) | 25.0 (12.4) | 27.6 (12.2) |
| Impairment at T2, M (SD) | 21.4 (10.3) | 19.3 (7.5) | 15.9 (6.7) | 22.2 (12.1) |
| Impairment at T3, M (SD) | - | - | 17.9 (10.2) | 21.9 (13.3) |

*Note*: OLP = open-label placebo; TAU = treatment as usual; T1 = baseline assessment; T2 = assessment ca. two weeks after the first assessment; T3 = assessment ca. two weeks after the second assessment (this assessment was completed by the TAU group only)

[1] Total score ranges from 0 to 18

[2] Total score ranges from 0 to 70

**Controlling for medication use.** We added medication use as a covariate in the above-mentioned Time by Treatment by Encounter mixed ANOVA to control for differences in medication use. This ANCOVA indicated that the main effect of Time was not significant any longer, $F(1, 49) = 0.293$; $p = .591$; $\eta^2_p = .006$. The other main and interaction effects remained non-significant. Interestingly, medication use had a significant effect on symptom improvement, $F(1, 49) = 4.360$; $p = .042$; $\eta^2_p = .082$. To further explore the effects of medication use on symptom improvement, we performed an additional medication by placebo ANOVA with the change score (symptoms T1 –symptoms T2) as the dependent variable, showing a significant

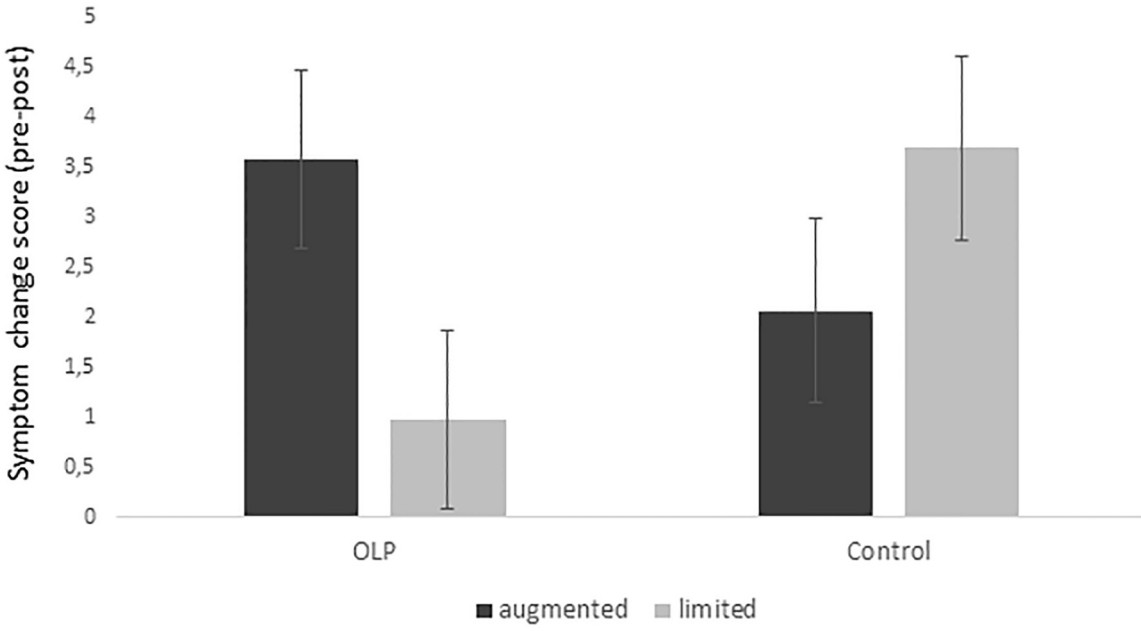

**Fig 4. Response to open-label placebo (OLP) vs. treatment as usual (control) as a function of the clinical encounter when controlling for baseline symptoms in an ANCOVA.**

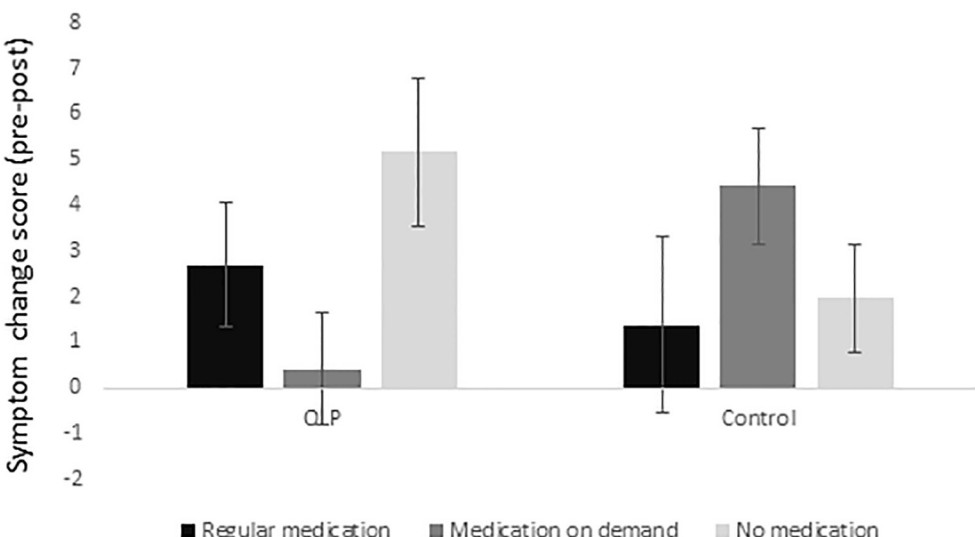

**Fig 5. Response to open-label placebo (OLP) vs. treatment as usual (control) as a function of medication use.**

medication by treatment interaction ($F(2, 48) = 3.385$; $p = .042$; $\eta^2_p = .124$): participants in the OLP group showed the largest symptom reduction if they did not take any medication against their allergy (adj. $M = 5.17$; $SE = 1.78$), whereas participants who took medication on demand hardly benefitted from OLP (adj. $M = 0.42$; $SE = 1.26$). In the control group, by contrast, symptom improvement was greatest if participants took medication on demand (adj. $M = 4.42$; $SE = 1.26$), as illustrated in Fig 5.

### Secondary endpoint: Change in impairment

The Time by Treatment by Encounter ANOVA with current impairment by allergic symptoms as the dependent variable indicated no significant interactions. There was a significant main effect of Time, $F(1, 50) = 14.056$; $p < .001$; $\eta^2_p = .219$, with overall higher symptom severity at baseline than at follow-up (for the descriptive values, see Table 4). The main effects of Treatment ($F(1, 50) = 0.008$; $p = .927$; $\eta^2_p < .001$) and Encounter ($F(1, 50) = 0.465$; $p = .499$; $\eta^2_p = .009$) were not significant.

As for the primary endpoint, we also conducted an ANCOVA controlling for baseline impairment, with the pre to post change in impairment as the dependent variable. This ANCOVA indicated no significant main effect of Treatment ($F(1, 49) = 0.419$; $p = .520$; $\eta_p^2 = .008$) and no significant main effect of the Encounter ($F(1, 49) = 0.545$; $p = .464$; $\eta_p^2 = .011$). The Treatment by Encounter interaction was not significant either, $F(1, 49) = 2.618$; $p = .112$; $\eta_p^2 = .051$.

Of note, for the secondary endpoint, medication use had no influence on the results, as indicated by an ANCOVA.

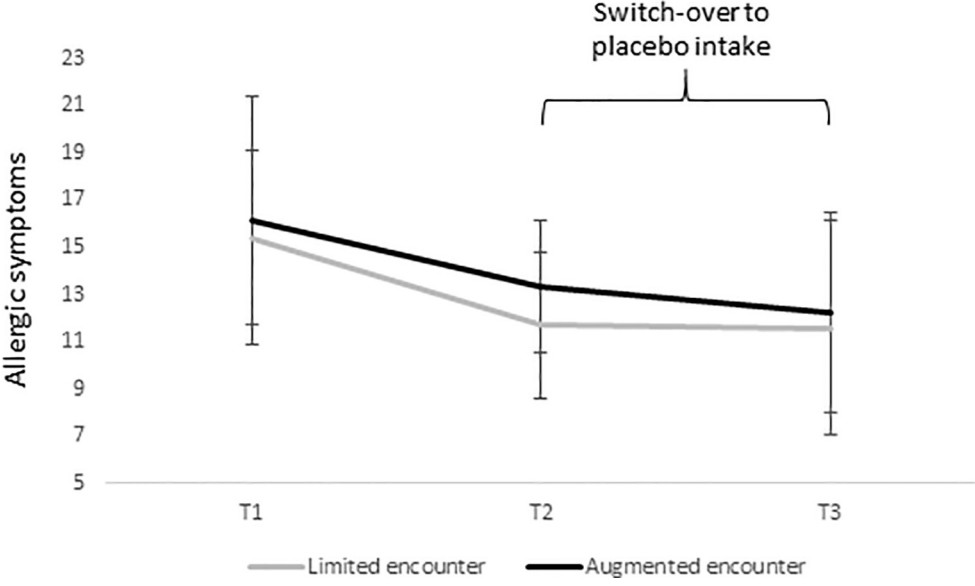

## Change in allergic symptoms in the control group

Fig 6. Symptom change in the control group after taking the placebo.

### Switch-over in the control group

We performed a 3 (Time: T1 vs. T2 vs. T3) by 2 (Encounter: augmented vs. limited) mixed ANOVA to examine the course of symptoms in the control group; in particular, we analyzed whether they benefitted from switching over to placebo intake after T2. The results showed no significant interaction, but a significant main effect of Time ($F(1.358, 20.376) = 5.782$; $p = .018$; $\eta^2_p = .278$); specifically, allergic symptoms decreased from T1 to T2 ($t(25) = 3.259$; $p = .003$), while there was no significant symptom reduction from T2 to T3 ($t(16) = 0.508$; $p = .618$), indicating that the intake of placebos after T2 did not additionally reduce allergic symptoms in the control group (as depicted in Fig 6). The main effect of the clinical encounter was not significant ($F(1, 15) = 0.546$; $p = .471$; $\eta^2_p = .035$).

### Beliefs about COVID-19

To take into account the special circumstances of our recruitment period, i.e., amid the COVID-19 pandemic, we asked participants whether they were concerned about the possibility that their symptoms could be related to COVID-19 (first item) and whether they believed their symptoms to be independent from COVID-19 (second item). We found that the belief that symptoms were unrelated to COVID-19, as assessed at T1, did not significantly correlate with changes in allergic symptoms from T1 to T2, $r = -.032$; $p = .873$; however, the same belief assessed at T2 did significantly correlate with symptom change from T1 to T2, $r = .338$; $p = .012$, indicating that the more confident participants were that their symptoms were not related to COVID-19, the larger the reduction in symptoms. Interestingly, the magnitude of the correlation between the aforementioned belief at T2 and symptom change differed considerably between OLP and the control group: in the OLP group, the correlation was small,

negative, and non-significant ($r$ = -.164; $p$ = .404), whereas a significant positive correlation was found for the control group ($r$ = .582; $p$ = .002). The opposite belief, expressing concerns about a relationship between current symptoms and COVID-19, did not correlate with symptom change (neither at T1, $r$ = .085; $p$ = .543, nor at T2, $r$ = .019; $p$ = .892), in none of the treatment groups.

## Patients' feedback

After completing the study, we asked participants for their experience of taking the placebo in semi-structured interview questions. The analysis of this data revealed that there was considerable heterogeneity in patients' responses to OLP: some reported to have benefitted from it enormously, whereas others mentioned that they did not notice any effects. Of those who said to have improved on OLP, some patients linked their improvement to the placebo, whereas others attributed their symptom reduction to other factors (such as changes of the weather). Interestingly, some patients mentioned to have experienced positive effects of the placebos on symptoms other than those assessed by the primary outcome measure (which was limited to nose- or eye-related symptoms). For instance, some patients explained that they had noticed less skin irritations and less fatigue after taking the placebo.

## Discussion

To our knowledge, the present study was the first to examine the effects of remotely provided non-deceptive placebos. Unlike previous studies on OLP in allergic rhinitis [16, 17], the current study did not provide evidence for the hypothesis that OLP reduces allergic symptoms relative to TAU. Aside from allergic rhinitis, the failure of OLP in the present study is also discrepant from previous studies showing beneficial effects of OLP on a variety of physical and mental conditions [5–9, 11, 12, 15, 34]. To explain these discrepancies, several explanations might be considered.

First, as our study was the first to investigate the effects of remotely administered OLPs, it appears most plausible that the remote provision might account for the failed replication of previous findings. If that interpretation turned out to be true, it could imply that a clinical encounter with physical contact between patient and provider is a prerequisite for OLPs to be effective. Alternatively, it also conceivable that the delay in taking the placebos after the encounter accounts for the non-significant effects of the OLP: Specifically, in previous studies, participants received the placebos immediately after the clinical encounter and started taking them accordingly; in our study, however, there was a delay of 2–3 days owing to the time the postal service needed to deliver the placebos. Thus, the idea of OLP as a promising treatment option as discussed with the provider might not have been salient enough any longer when participants started to take the placebos. Second, it might be that the markedly different recruitment circumstances during the COVID-19 pandemic influenced the results. Possibly, unlike participants from previous OLP studies, participants from the current study had serious issues aside from their allergic symptoms to deal with, which may have led them to be less sensitive to possible effects of the placebo. Whatever the exact influence, the remarkably different correlation between beliefs about COVID-19 and symptom change in the two treatment groups suggests that the pandemic might have affected the results in some way. Third, in comparison to the two previous studies examining the effects of OLP specifically on allergic rhinitis [16, 17], the present study used a different questionnaire to assess allergic symptoms, following recent expert recommendations [28]. In addition to the remote provision of placebos, this represents a second difference from prior work; therefore, it is possible that the narrower focus of the questionnaire used in the present study also accounts for the discrepant results to some

extent. In fact, some patients reported in the interview after the completion of the study that they had noticed positive effects of the placebo that were not covered by the symptom questionnaire of the present study, unlike the questionnaire used by Schäfer et al. [16, 17].

Interestingly, additional analyses suggested that medication use may influence the response to OLP vs. TAU. Specifically, we found that participants hardly reported any effects of the OLP when they took medication on demand, whereas they did improve quite strongly on OLP when they did not take any medication for their allergic symptoms. In the control group, on the other hand, symptom reduction was greatest when participants took medication on demand. These results should be interpreted with caution given the small sample size and the low numbers of participants in each cell; yet, these findings may raise the interesting question of how medication intake influences the response to OLP vs. TAU.

An additional aim of the present study was to investigate the role of the clinical encounter in OLP treatment. Owing to the inevitable modification of the variation of the clinical encounter due to the restriction of physical contact in relation to the pandemic, we were unsure initially about whether varying the encounter in a virtual setting is possible at all. In this respect, the successful manipulation check of the current study is encouraging as it shows that even in a remote study, it is possible to manipulate the perception of a clinical encounter. Moreover, although the clinical encounter did not influence symptom reduction in the main analysis, it did have an influence in the ANCOVA controlling for baseline levels of allergic symptoms. In particular, we found a significant interaction between the treatment received (OLP+TAU vs. TAU alone) and the augmented vs. limited encounter, in the sense that participants from the OLP group showed more symptom improvement when they received an augmented clinical encounter, whereas participants from the control group improved more in the limited encounter condition. In other words, if participants were prescribed OLP, they benefitted from it more if it was provided by a warm, empathic provider, whereas TAU was more effective when participants underwent a rather neutral, un-personalized, and distant encounter. To interpret this unexpected finding, the following points might be considered.

First, the significant interaction might reflect some kind of disappointment of the TAU group in the augmented condition in response to the information that they were randomized to the control group. Conceivably, the warm and friendly behavior of the provider in the augmented condition might have raised the expectation of being prescribed an effective treatment, which might then have led to disappointment when being informed that one would receive the placebos only after the second appointment two weeks later. In the limited encounter, participants may not have been that disappointed about being randomized to the control group as the outcome of the encounter (staying with TAU) was more consistent with the provider's behavior. Second, on similar lines, participants who received placebos after the augmented condition may have felt some sort of desire to please the warm and empathic provider, potentially resulting in demand effects regarding symptom development in the augmented OLP condition.

## Implications for future work

In our view, merit of the present study can be seen in the questions it raises. For instance, our study demonstrated that providing OLPs remotely is possible, which might be an important conclusion itself for future placebo studies, given the unpredictable further duration of the current pandemic (and its consequences for doing research); however, according to our findings, it might be that OLPs are less effective when provided remotely (for various reasons, as discussed above). Since the remote conduction of the present study was not the only aspect in which it differed from previous work, that conclusion cannot be drawn with certainty, though.

Thus, future work might evaluate the potential of remote OLPs by equaling the procedure of previous studies in all respects except for providing placebos remotely, as opposed to providing them as part of the physical encounter. Indeed, a recent study from another area of research suggests that performing the same experiment in the laboratory vs. remotely may not lead to the same results [35].

Further implications might be derived from the fact that our study was the first to manipulate the clinical encounter to examine it as a potential factor contributing to the response to OLP. While our study shows that it is possible to manipulate the clinical encounter in a remote study, it is less clear how—and why—the clinical encounter influences the response to OLP vs. TAU. In particular, whereas the clinical encounter did not affect symptom change in the main analysis, it did have an effect when controlling for baseline symptoms. Although this interaction effect raises interesting questions for future research by pointing to possibly differential effects of an augmented vs. limited encounter on symptom perception for OLP vs. TAU, the robustness of this effect still needs to be further investigated before drawing conclusions about it.

## Strengths and limitations

Strengths of our study can be seen in the investigation of the provision of OLP remotely for the first time; the conduction of a pilot study to pre-examine the perception of the variations of the clinical encounter; the successful manipulation check with respect to the clinical encounter; the standardized procedure including a protocol for the provider's behavior and the randomized assignment of participants to the respective conditions; the consideration of patients' beliefs about the current COVID-19 pandemic in relation to their symptoms; and the systematic assessment of participants' treatment expectancies as well as their understanding of placebos. Notwithstanding these strengths, the present study also has several limitations that need to be considered.

The most significant limitation is the small sample size ($N = 54$) with less than 30 participants per treatment group and less than 15 persons in each of the four arms, thus limiting the chance to uncover potential differences between the groups. Yet, it should be noted that previous OLP studies examining people with allergic rhinitis had even smaller sample sizes (i.e. $N = 25$ in [16] and $N = 46$ in [17]), and based on the large effect sizes revealed in that prior work, the a-priori power analysis indicated a minimum sample size of 52 participants, which was reached, actually. Nevertheless, we acknowledge that the power analysis was based on the original plan to provide OLPs in a regular clinical encounter with physical contact, and under the assumption that OLPs are less effective when provided remotely, the current sample size might not have been large enough to detect significant effects of the placebo over TAU. Furthermore, in our attempt to follow expert consensus regarding the unification of primary endpoint measures in studies on allergic rhinitis [28], we used a questionnaire different from the one used in previous work [16, 17], which makes the comparison of our results with prior research more difficult. Also, it would have been valuable to have data on participants' allergic reactions beyond self-report data, e.g., more physiological and immunological data. Moreover, it would have been interesting to assess the effects of the augmented vs. limited encounter on additional psychological variables, such as negative affect [36], to get a broader understanding of the effects of the encounter and to see whether such psychological variables could have influenced the effects of the placebo.

## Conclusions

Owing to the dramatic consequences the COVID-19 pandemic has had for various areas of life, including research, the present study was the first to evaluate the potential of providing

non-deceptive placebos remotely. The study demonstrates that the remote provision of OLP is feasible, yet their effectiveness might be lower than those found in previous research for OLP provided within a physical encounter. In addition to the main finding regarding the overall effects of OLP relative to TAU, the present study provided some interesting additional findings concerning the interaction of OLP with the clinical encounter and medication use. Thus, the current study raises several questions for future research about the potential and limits of OLP.

## Supporting information

**S1 Checklist. CONSORT 2010 checklist of information to include when reporting a randomised trial**[*]**.**
(DOC)

**S1 Table. Items of the extended health screening experience questionnaire.**
(DOCX)

**S1 File.**
(PDF)

**S2 File.**
(PDF)

## Acknowledgments

We are very grateful to Michael Schäfer for his helpful advices in planning the study and his support of our project. In addition, we would like to thank the "Adler Apotheke", the local pharmacy in Landau, Germany, for their support in producing the placebos used in the present study.

## Author Contributions

**Conceptualization:** Tobias Kube, Irving Kirsch.

**Data curation:** Tobias Kube, Verena E. Hofmann.

**Formal analysis:** Tobias Kube.

**Investigation:** Tobias Kube, Verena E. Hofmann.

**Methodology:** Tobias Kube.

**Project administration:** Tobias Kube, Verena E. Hofmann.

**Supervision:** Julia A. Glombiewski, Irving Kirsch.

**Visualization:** Tobias Kube, Verena E. Hofmann.

**Writing – original draft:** Tobias Kube.

**Writing – review & editing:** Tobias Kube, Verena E. Hofmann, Julia A. Glombiewski, Irving Kirsch.

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
