## [Decision Letter · Decision Letter 0]

12 Jan 2021

PONE-D-20-34179

Providing Open-Label Placebos Remotely – A Randomized Controlled Trial in Allergic Rhinitis

PLOS ONE

Dear Dr. Kube,

Thank you for submitting your manuscript to PLOS ONE. After careful consideration, we feel that it has merit but does not fully meet PLOS ONE’s publication criteria as it currently stands. Therefore, we invite you to submit a revised version of the manuscript that addresses the points raised during the review process.

We look forward to receiving your revised manuscript.

Kind regards,

Zheng Liu, M.D., Ph.D.

Academic Editor

PLOS ONE

Journal Requirements:

2. Please include captions for your Supporting Information files at the end of your manuscript, and update any in-text citations to match accordingly. Please see our Supporting Information guidelines for more information: http://journals.plos.org/plosone/s/supporting-information

Reviewers' comments:

Reviewer's Responses to Questions

**Comments to the Author**

1. Is the manuscript technically sound, and do the data support the conclusions?

Reviewer #1: Partly

Reviewer #2: Yes

Reviewer #3: Yes

2. Has the statistical analysis been performed appropriately and rigorously? 

Reviewer #1: Yes

Reviewer #2: Yes

Reviewer #3: Yes

3. Have the authors made all data underlying the findings in their manuscript fully available?

Reviewer #1: Yes

Reviewer #2: Yes

Reviewer #3: Yes

4. Is the manuscript presented in an intelligible fashion and written in standard English?

Reviewer #1: Yes

Reviewer #2: Yes

Reviewer #3: Yes

5. Review Comments to the Author

Reviewer #1: The authors investigated the placebo effect in an allergic rhinitis randomized controlled trial by providing placebo remotely and virtual clinical encounter. They concluded it was possible to provide open-label placebo and enhance the encounter remotely, but the effectiveness might be lower than physical patient-provider interaction. In general, this is a well-designed study, the authors described their findings with details. The study is of novelty, especially in the COVID-19 pandemic period. My major concern is the small sample size of this study, as only 54 patients were enrolled in this 4-arm clinical trial and an average of 14 cases in each arm. I’m not sure if it is qualified to discriminate the potential differences among the 4 subgroups, although the authors have acknowledged it as a limitation in the discussion. Meanwhile, the manuscript can be shortened and avoid narrating repeatedly.

Reviewer #2: The authors presented the interaction between OLP (with and without TAU) and encounter (augmented and limited) during COVID-19 pandemic, and found that the remote provision of OLP is feasible, but their effectiveness might be lower than those found in previous research for OLP provided within a physical encounter. There are several questions,

1.Augmented and limited encounters may have some impact on patients’ psychology, do you think there are some impact on the effects, especially in control & limited groups?

2.Previous studies have showed that network may help patients to use medicine regularly, and benefit to symptoms control. However, in this study, the authors have different results. I want to know that in no treatment subgroup, did the patients have the TAU or OLP regularly? If they took the medicine (OLP or TAU) as other subgroups, changes of symptom score may origin from the higher score in no treatment group and the lower score in regular treatment group.

3.What’s the frequency of regular treatment, treatment on demand and no treatment in four subgroups of OLP-augmented, OLP-limited, TAU-augmented, and TAU-limited? Do you think there are some impact on the results of the four subgroups respectively?

Reviewer #3: A two-arm placebo controlled randomized clinical trial was conducted to compare TAU alone to OLP+TAU and clinical encounter (augmented vs. limited). Primarily ANOVA models were used to test for differences in outcomes between treatments and clinical encounters. Participants from all treatment groups showed significant symptom reduction from baseline to two weeks later, but OLP had no incremental effect over TAU.

Minor revision:

1- Indicate the statistical testing method used in G*Power to justify the sample size.

2- Indicate if the following procedure was followed when testing main and interaction effects. If the interaction effect is significant, provide an interpretation of the results, but do not test main effects because the tests for main effects are uninteresting in light of significant interactions. If interaction effects are non-significant, drop the interaction effects from the model and test the main effects. Determining which results to present when testing interactions is often a multi-step process.

6. PLOS authors have the option to publish the peer review history of their article (what does this mean?). If published, this will include your full peer review and any attached files.

Reviewer #1: No

Reviewer #2: No

Reviewer #3: No

---

## [Author Response · Author response to Decision Letter 0]

24 Jan 2021

Please find our detailed responses to the reviewers in the file labelled "Response to Reviewers".

---

## [Decision Letter · Decision Letter 1]

25 Feb 2021

Providing Open-Label Placebos Remotely – A Randomized Controlled Trial in Allergic Rhinitis

PONE-D-20-34179R1

Dear Dr. Kube,

We’re pleased to inform you that your manuscript has been judged scientifically suitable for publication and will be formally accepted for publication once it meets all outstanding technical requirements.

Kind regards,

Zheng Liu, M.D., Ph.D.

Academic Editor

PLOS ONE

Additional Editor Comments (optional):

Reviewers' comments:

Reviewer's Responses to Questions

**Comments to the Author**

1. If the authors have adequately addressed your comments raised in a previous round of review and you feel that this manuscript is now acceptable for publication, you may indicate that here to bypass the “Comments to the Author” section, enter your conflict of interest statement in the “Confidential to Editor” section, and submit your "Accept" recommendation.

Reviewer #1: All comments have been addressed

Reviewer #3: All comments have been addressed

2. Is the manuscript technically sound, and do the data support the conclusions?

Reviewer #1: Yes

Reviewer #3: (No Response)

3. Has the statistical analysis been performed appropriately and rigorously? 

Reviewer #1: Yes

Reviewer #3: (No Response)

4. Have the authors made all data underlying the findings in their manuscript fully available?

Reviewer #1: Yes

Reviewer #3: (No Response)

5. Is the manuscript presented in an intelligible fashion and written in standard English?

Reviewer #1: Yes

Reviewer #3: (No Response)

6. Review Comments to the Author

Reviewer #1: Although the sample size is relatively small in this study, the authors discussed this issue thoroughly and noted it as the major limitation in the revision manuscript. I have no further comments.

Reviewer #3: (No Response)

7. PLOS authors have the option to publish the peer review history of their article (what does this mean?). If published, this will include your full peer review and any attached files.

Reviewer #1: No

Reviewer #3: No

---

## [Editor Report · Acceptance letter]

2 Mar 2021

PONE-D-20-34179R1 

Providing Open-Label Placebos Remotely – A Randomized Controlled Trial in Allergic Rhinitis 

Dear Dr. Kube:

I'm pleased to inform you that your manuscript has been deemed suitable for publication in PLOS ONE. Congratulations! Your manuscript is now with our production department. 

Kind regards, 

on behalf of

Dr. Zheng Liu 

Academic Editor

PLOS ONE